Alethia Fernández de la Reguera Ahedo

Institute of Legal Research

National Autonomous University of Mexico

January 15, 2024.

## Punitive subjectivities and emotions in immigration detention

**Abstract**

In immigration detention centres emotions run high; tensions, conflicts, anxiety, and affection occur amid bureaucratic procedures, paperwork, files, lists of people, systematization of cases, buses that come and go with detainees, and people who will be deported. Although immigration detention belongs to the realm of administrative law, in practice, detention centres operate closer to a penal detention. Little is known about the operation of immigration detention centres in Mexico, including how punishment takes place in a daily practice. Even less is known about the people who work there, especially how they collaborate with a system that promotes punishment as a central element of immigration detention.

In this article, I study how emotions such as fear and disgust can enhance a punitive subjectivity of *Instituto Nacional de Migración* [National Institute of Migration] (INM) agents, while empathy can challenge it. I analyse working conditions and daily interactions in detention centres, immigration control facilities and their surroundings, all of them environments of discretion and social distancing between INM officers and migrants. I argue that punishment becomes central as an institutionalised practice in these places by channelling emotions of INM agents derived from anxieties and frustrations and promoting a sense of institutional belonging and the illusion of order and control.

However, I also analyse empathy, which even if it´s fragile, shows the existence of other subjectivities within the punitive system of immigration detention. In this sense, the questions I intend to answer in this paper are: Under what structural and institutional conditions do emotions become power in immigration detention settings? What impact do emotions have on the construction of a punitive subjectivity? How do emotions expressed by INM agents (such as fear, disgust and compassion) enhance or challenge punishment of migrants?

**Introduction**

In 2017, I met Angélica[1], a 26-year-old Honduran woman detained for about a month at the *Estación Migratoria Siglo XXI* [XXIst Century Immigration Station]. The day I met her, I entered as part of the psycho-legal team of the organisation Fray Matías de Córdova Human Rights Centre The visit was during the usual hours of 10 am to 1 pm and the aim was to follow-up and identify new cases of people who required psycho-legal accompaniment in detention, such as asylum seekers, families, people with illnesses, pregnant women, people who needed translation services or cases of human rights violations during detention. The interview with Angelica took place in a tiny, airless office. I was uncomfortable not being able to provide her with a dignified space, not even a glass of water. Offering her a chair instead of sitting on the floor was the best we could do.

It was evident that she was under a great deal of stress and anxiety. She had fled from a violent partner, had left her two children in Honduras in the care of her mother and had been detained in the city of Tuxtla Gutiérrez and then taken to *Estación Migratoria Siglo XXI* the same day she had entered Mexican territory. She narrated that in the last few nights, she had had panic attacks and a lot of difficulty falling asleep. She could not breathe as she recounted her confusion at not understanding why she was being detained or how much longer the detention would continue. In recent days, her blood pressure had risen. She was afraid to complain to the authorities and felt very lonely. She had not been menstruating for several weeks and had not had access to a pregnancy test. After the interview, we managed to get her access to the test, which came back negative. She, like many women in immigration detention, had stopped menstruating due to the emotional stress of being deprived of her liberty.

Weeks after the meeting with Angélica, I was able to interview Nancy, a former agent of the *Instituto Nacional de Migración* [National Migration Institute] (INM) who had worked for four years as an agent in the Saltillo immigration station[2] in northern Mexico, almost 2,000 kilometres from Tapachula´s *Estación Migratoria Siglo XXI*. Like Angélica, she had also stopped menstruating for three months due to the emotional stress of working

---

[1] I changed the names to respect the anonymity of the interviewees.
[2] In this article, I use the terms immigration station and immigration detention centre as synonyms. While the terms used in the Migration Law are immigration station or temporary stay, in practice, they are migrant prisons or immigration detention centres.

there. "I was in emotional pain," she said. Nancy became ill with gastritis, and her hormonal system was affected. Like Angélica, she had very high blood pressure.

What happened to both of them? Why did they stop menstruating? Why did they have these symptoms? For very different reasons, they both spent too many hours, days and weeks in an immigration detention centre. Angélica was deprived of her liberty with the uncertainty of not knowing why she had been detained, how long her detention would last and how she could get out. Nancy worked for four years in this place. She was hired even before she graduated from university. She entered the INM with great motivation, but after a while, she was moved to another area within the same office and began to suffer mistreatment by her boss and sexual harassment by a colleague. The trauma they experienced at the immigration detention centre, in different contexts and for other reasons, had an impact on their physical and mental health.

In the past years, I have studied power relations in detention centres and the impact of the subjectivities of immigration agents on the implementation of immigration policy (Fernández de la Reguera, 2020, 2022). In the testimonies I have obtained through interviews with migrants during and after having been detained, as well as with some people who work or have worked in the INM (in regulation offices or immigration stations), I have identified that trauma remains and has effects beyond the detention stage or quitting their jobs, and often causes harm not only in migrants but also in the immigration bureaucrats. Two former female agents I have interviewed, after resigning, went through a long process of recovery and healing, like Nancy who left Mexico for a year and eventually returned to work in the public sector, but in a very different area. These stories made me realise that on the one hand, the experience of being (detained or working) in a migrant prison is deeply emotional, and on the other hand anti-migrant populist emotions are institutionalised in detention centres (Bilgic & Gkouti, 2021).

Since 9/11, governments in the Global North have prioritized immigration detention to demonstrate that they have control of the borders. It is a policy that "[...] has emerged as a crucial element of the search for a robust and credible system" (Hall, 2010, p. 882). Mexico has not been left behind and has one of the largest immigration detention systems in the world, with about 50 centers for the deprivation of liberty of persons in mobility (about 37 immigration detention centers[3] with an average stay of 3 weeks and 11 temporary

---

[3] According to Art. 3 of the 2011 Migration Law, an immigration station is a physical facility established by the National Migration Institute to "temporarily house foreigners who cannot prove their regular migratory status" while their migratory status is resolved.

detention centers with a maximum stay that ranges from 48 hours to seven days (Global Detention, 2021).  In strict sense detention periods should comply with administrative detention regulations, which stipulate a maximum period of 36 hours. However, the Migration Law provides for a detention period of 15 working days, which can be extended to 60 working days. Moreover, Art. 111 stipulates that if migrants file an administrative or judicial appeal to claim issues inherent to their immigration status in the national territory, the immigration detention periods are suspended indefinitely.

Between January and November 2023, 686,732 irregular migrants were detained of which 486,424 were taken to a detention centre (Unidad de Política Migratoria, 2023). Unfortunately, Mexico fulfils all the characteristics of arbitrary detention (CPDTMF, 2021) since people are generally and systematically detained, including vulnerable populations such as pregnant women, the elderly, children and adolescents and the sick. Despite legislation prohibiting the detention of accompanied and unaccompanied children and adolescents, it is even possible to document cases of these populations in immigration stations.

Although immigration detention belongs to the realm of administrative law, in practice, detention centres operate closer to a penal detention, which is why they are increasingly the subject of study by criminology (Weber, 2005). Little is known about the operation of immigration detention centres in Mexico, including how punishment takes place in a daily practice such as: denying information, forbidding a phone call, denying access to medicine, a clean toilet, a glass of water or a sanitary towel. Even less is known about the people who work there, especially how they collaborate with a system that promotes punishment as a central element of immigration detention.

In this article, I study how emotions such as fear and disgust can enhance a punitive subjectivity of INM agents, while empathy can challenge it. I analyse working conditions and daily interactions between INM officers and migrants in detention centres, immigration control facilities and their surroundings. I argue that punishment becomes central as an institutionalised practice in these environments by channelling emotions of INM agents derived from the anxieties and frustrations about their working environment. Moreover, punishment allows them to have a sense of belonging and social differentiation from the migrants and brings them the illusion of order and control (Carvalho & Chamberlen, 2024).

However, there is also empathy, which even if it´s fragile, shows the existence of other subjectivities within the punitive system of immigration detention. In this sense, the

questions I intend to answer in this paper are: Under what structural and institutional conditions do emotions become power in immigration detention settings? What impact do emotions have on the construction of a punitive subjectivity? How do emotions expressed by INM agents (such as fear, disgust and compassion) enhance or challenge punishment of migrants?

The article has four sections, plus the introduction and conclusion. In the first section, I point out some of the most relevant research in the study of migration policies from the perspective of emotions, especially the role of emotions in the institutional processes of immigration detention. In the second section, I present central concepts for my analysis, such as Sara Ahmed's affective economies, to understand how emotions are significant elements of power dynamics and punishment in immigration detention. In this section, I also explain the link between institutional ethnography and sociology of emotions to understand what this approach allowed me to explore at the *Estación Migratoria Siglo XXI*. In the third section, based on my empirical analysis I present some reflections on the punitive subjectivity of the INM´s agents, especially the internal and external context in which they work, to understand how the precariousness of work and the risks associated with organised crime affect the agents' punitive subjectivity. In the last section I analyse three emotions: fear, disgust and empathy in relation to the construction of a punitive subjectivity that generates an institutional identity, a sense of belonging and an affective disposition to punish.

**Understanding the impact of emotions in immigration policies**

In Latin America, the field of study of migrations and emotions is in process of consolidation (see Ariza, 2021). Research has mainly focused on the perspective of the migrant subject, for example, on the effects that emotions have on the migratory project of people (Herrera & Rivera, 2021). In other latitudes, research linking emotions with immigration policies is becoming a prolific field of study (Graham, 2002; Moss & Prince, 2017; Savio Vammen & Sy Syro Rivera, 2021; Tessitore, Caffieri, Parola, Cozzolino, & Margherita, 2023). Previous research has raised substantive questions for understanding the analytical potential that emotions bring to studying policies, institutions and subjects in charge of implementing immigration policies and detention. For example, the American sociologist Irene Vega (2017) shows how while suppressing certain emotions and surfacing others, immigration agents constantly debate between rationality and emotion.

My interest is linked to research that has focused on how States instrumentalize affective technologies of power to control and punish asylum seekers, who often experience shame, pain, stress, and fear (Meier, 2020); and on the institutionalization of emotions to demonstrate that negative emotions such as anxiety, fear, and anger are manifested in concrete policies and actions, especially to generate emotional effects on female asylum seekers in detention (Bilgic & Gkouti, 2021).

The British geographer Isabel Meier (2020) analyses under what conditions affect and emotion become power and violence. She speaks of emotional borderwork to explain the amount of time and energy that takes to cross borders and access international protection; but also about discomfort and precariousness as conditions that states impose on migrants, who end up overwhelmed and with mental distress. In this sense, affective border violence works subtly and gradually through emotions that disempower, and subordinate migrants to state control.

However, emotional harm is difficult to assess, especially as part of the institutional practices of bordering. The British social scientist Melanie Griffiths studies the emotional management at the institutional and individual level of staff working for the UK Immigration system, finding that the four most common emotions are: anger, disgust, suspicion and fear. She observes how in policy and practice "[…] emotions and encounters are embedded in inescapable power dynamics" (2023, p. 6). In different formats crossed by gender, race and social class, emotional interactions between migrants and the UK immigration agents, seem to determine access or denial of rights and procedures. For example, anger or disgust can enhance suspicion affecting the way officers interpret the evidence provided by asylum seekers.

For her part, Argentine criminologist Ana Aliverti (2021) shows that immigration agents work with constant moral issues between a humanitarian and a punitive approach. Since their work demands emotional labour and constant ethical dilemmas of care and control, emotions are key to understanding the decision-making processes of street-level bureaucrats in charge of border enforcement in the UK. She finds that although many officers employ indifference and detachment as a way of coping with their moral dilemmas, there are also officers who express compassion and despair when dealing with cases involving families and children. She further demonstrates the impact of operational stressors, primarily work overload and emotional stress, that usually go unnoticed by the institution. She concludes: "In conciliating the conflicting demands for care and order, empathy and suspicion, these officers often felt unable to achieve either" (2021, p. 152).

Working in immigration detention centres demands establishing social and emotional barriers with incarcerated migrants, to avoid emotional and ethical conflicts (Puthoopparambil et al., 2015). The work of British geographer Alexandra Hall demonstrates the importance of placing emotions at the center of the analysis of power relations and the functioning of a detention center. "For the study of places like Lockson (UK immigration detention center), where emotions are bound up with the working center, such an approach has value for taking seriously the ways in which emotion merges with, sustains, and creates a sense of difference" (Hall, 2010, p. 886).

An apparent solution for some of these bureaucrats to emotionally detach from their work is to act with indifference. In their research, Weber & Landman (2002) asked immigration officers in the UK if they ever wondered about the people they detained, such as how long they would be detained or what happened to them. Half responded that they did not due to lack of time or having to attend to other cases. Other guards responded that they do not because they keep a distance between their life and their work. Once they leave their office, they forget about work issues. "Organizational actors may indeed be personally absent from decision-making processes of which they are part and may be emotionally indifferent to, genuinely unaware of, or otherwise linked to, the harmful consequences of their actions" (Weber, 2005, p. 91). In any case, emotional management is an intrinsic part of their work. It is a vein that requires further analysis in the study of bureaucracies and the implementation of immigration policy.

**Emotions and institutional ethnography to study immigration detention in Mexico**

Since 2017, I have intermittently carried out fieldwork in detention facilities[4], especially in and around Tapachula, a border city adjacent to Guatemala with the largest detention center in the country, with a capacity to detain 960 people. In addition, this city is the last point before deportation of people detained all around the country and the first immigration detention place for new arrivals to Mexico from the southern border. I chose to do my research at the *Estación Migratoria Siglo XXI* in this city due not only to its size

---

[4] As an academic access to immigration detention centres in Mexico is often minimal, as it is necessary to obtain a permit through civil society organisations that provide psycho-legal accompaniment in these places. I have carried out this research thanks to the support and collaboration I have established with various civil society organisations.

but also to the diversity of profiles of people in mobility, and the growing militarization of the area.

Tapachula is a city that has changed, especially in the last decade, by being not only a transit city, but a city of deportation and detention, impacting not only migrants but also the city and those who live there. Moreover, it has a colonial history of exploitation of indigenous labour from Mexico and Guatemala in the coffee-growing areas, which has left behind structures of racism, violence and discrimination, today fuelled by the criminalisation of migrants. In this research, I observed what feminist geographer Alison Mountz (2017) calls affective eruptions or how trauma is transmitted and expressed in immigration detention centres and detention settings. These places belong to a wider detention industry that exploits local communities where the facilities are allocated, spreading trauma from a colonial past to the present.

Mountz explains that "[…] the despair and feeling of powerlessness of a manager with power over others signals the historical grounds on which detention facilities are built and civil servants required to work as "the bitch" of the federal government in its latest project to contain human mobility on the island" (Mountz, 2017, p. 80). In my research, I have identified that even though the INM agents are from Tapachula and are culturally and ethnically similar to people from Guatemala, they tend to be more racist than in other zones of the country.

Immigration detention centres are workplaces that work through punishment not only for the detained migrants but also for the staff in charge of custody and administrative procedures. I identify what the British criminologist Elaine Crawley (2013) describes as the culture of bullying in prisons, which occurs when staff who are disliked by their superiors are humiliated in front of the staff as a form of punishment. Furthermore, in the INM, there is ample scope for discretion. While this is not necessarily negative, as it has been studied that discretion in prisons can allow for the establishment of positive relationships and more flexible rules (Bennett, 2016), in this case, discretion at all levels of command provides power that can be used as a form of reward or punishment.

Although an immigration detention centre is not a prison as it falls under administrative law, the INM is a national security institution, and detention centres are very similar to prisons. In this sense, much of the literature on prisons is in line with my research findings. British criminologist Joe Sim explains how prison officers have an essential role in making prisons places of punishment. "Prison officers play their part in that process, not only through the politics of containment but more crucially, through the hegemonic

construction and objectification of the prisoner as the ultimate and only source of criminality in society" (Sim, 2012,p. 195); in this case the objectification of the migrant as a national threat. Within this context I address the existence of what criminologists Anastasia Chamberlen and Henrique Carvalho (2024) call punitive subjectivities or how immigration agents incorporate ways to show discipline, which for some is a synonym of punishment.

Moreover, the most punitive social contexts are those characterised by anxiety and a sense of crisis (Carvalho, H., Chamberlen, A., & Lewis, R., 2020), such as the public discourse on immigration management in Mexico, spercially in the south. We can even say that punishment begins long before the person is taken to a detention centre, for example, through racial profiling practices when they travel on buses, arrive at an airport and are arbitrarily detained by the authorities. In the south of Mexico, it is common for the authorities not to recognise or destroy identity documents. In fact, criminalisation of migrants usually establishes the norms creating a preexisting emotional state even before a person commits a crime or in this case an administrative offence by accessing the Mexican territory irregularly.

The institutional ethnography I carried out between 2017 and 2019 at *Estación Migratoria Siglo XXI* included around 12 weeks of participant observation and more than fifty in-depth interviews with (active and former) immigration agents and officers, immigrants, governmental officials, and staff from international organizations and NGOs[5]. It was not easy to find former officers to interview. It was through personal contacts and my work as an academic giving workshops on gender issues for various state and local government agencies that I was able to establish contacts.

Participant observation consisted of entering this place with the Fray Matías de Córdova Human Rights Centre staff to support their work of psycho-legal accompaniment of the detained population. Together with the team, I was in charge of preparing the visit and reviewing the NGO databases to verify the needs of each case. Once inside the detention centre, I accompanied migrants in their transfers and conducted interviews.

---

[5] Along with participant observation in this place, I carried out 14 interviews at the offices of the National Immigration Institute in Tapachula, Mexico City, and Tijuana; more than 30 interviews with migrants in detention, three interviews with female former public officials, six interviews with staff from different agencies, such as *Comisión Mexicana de Ayuda a Refugiados* [Mexican Commission for Refugees], *Comisión Nacional de Derechos Humanos* [National Commission for Human Rights], *Comisión Estatal de Derechos Humanos de Chiapas* [Chiapas State Commission of Human Rights], and Office of the United Nations High Commissioner for Refugees.

During the visits, I had the opportunity to have many informal chats with the guards and observe the administrative processes and the interaction between the guards, INM agents and the migrants. I have also made several observation visits to the vicinity of this place, where I have observed clashes between authorities and migrants.

Once I finished the first stage of the research in 2019, and after the pandemic, that is, from 2021 onwards, I kept visiting detention settings in Chiapas. These are places in very precarious conditions where people are in risk of being detained by INM, the National Guard and even local police, such as checkpoints or informal waiting places (public areas, parks and camps) where people stay for days or weeks to have access to transit or regularisation paperwork.

I depart from sociology of emotions as an approach to the social nature of emotions as well as the emotional dimension of social phenomena (Ariza, 2016). I understand emotions as experiences that arise through social interaction. At the same time, I recognize that "[...] emotions are 'subjective experiences that also have physiological, intersubjective, and cultural components'" (Crawford, 2014, p. 537). I analyse emotions as the result of social interaction and as a mediation between the subject and the sociocultural context.

British philosopher Sara Ahmed posits the existence of affective economies to emphasize that emotions have real effects by socially and politically situating individuals in their communities. She emphasizes the relationship between the psychic and the social, the individual and the collective, to understand how emotions work and produce an effect as they circulate both in social space and at the psychic level. "Affect does not reside in an object or sign, but is an effect of the circulation between objects and signs." (Ahmed, 2004, p. 120).

In a punitive social context, focused on punishment, migrants are criminalised through a public discourse that generates subjective and individual emotions such as fear, disgust, anger and compassion towards migrants. Therefore, what I explore is the formation of punitive subjectivities of immigration agents through the circulation of emotions. Which emotions are most commonly expressed by immigration agents? Which institutional rules and stressors allow officers to express certain emotions and hinder others? Under what structural and institutional conditions, do emotions become power in immigration detention settings? What impact do emotions have on the construction of a punitive subjectivity?

Ahmed understands borders as a consequence of affective economies in which affections circulate between objects and signs, and through the narrative of injury, the others are transformed into "the hated." Her theory shows that emotions align some subjects with others and against others. "[...] they create the very effect of the surfaces or boundaries of bodies and worlds" (Ahmed, 2004, p. 117). Although the focus of the article is not to analyse the discourses of hate surrounding migration, I do think it is important to point out that hatred is materialized in the body of migrants or those considered "the others" and places the hating subject (in this case the people who work in immigration detention centers) in the position of victims. In the years that I have conducted research in Tapachula, Chiapas, I have detected that the discourses of criminalisation of migrants not only generate rejection and dehumanization of people but also victimization of those in charge of controlling the borders.

I frame the analysis in an institutional ethnography to understand how emotions circulate as part of INM´s institutional setting, norms and culture. Previous research demonstrates that incorporating emotions as a source of knowledge in institutional ethnography significantly broadens the analytical lens of emotions as mediators between the subject and social action in institutional settings (Aliverti, 2021; Griffith, 2023; Hall, 2010; Weber & Landman 2002). This approach prioritizes posing questions arising from the tensions and contradictions observed (in a prolonged manner) in social interactions and that, in a latent manner, are present in the everydayness of institutional practices (Smith, 2001). "Institutional ethnography aims to explore and explain the social relations that organize experiences in institutional settings or settings in which these relations exist" (Kearney, Corman, Hart, Johnston, & Gormley, 2019, p.19). It studies subjects as social actors whose practices occur within explicit and implicit institutional rules.

This type of ethnography prioritizes the subjects who participate in the institutions. In this sense, it breaks with traditional sociology that concentrates on studying institutions from their organic functioning and not from the subjectivities of those who make up an institution. The institution is studied from the everyday experiences of its members. This approach allowed me to observe that emotions such as fear and disgust have an institutional function related to the centrality of punishment as a daily practice. They support on one hand, a sense of belonging, and on the other hand distancing from incarcerated migrants. On the contrary, empathy expressed by the INM agents toward migrants can become a form of institutional challenge of punishment.

In detention centres emotions run high; tensions, conflicts, anxiety, and affection occur amid bureaucratic procedures, paperwork, files, lists of people, systematization of cases, buses that come and go with detainees, and people who will be deported. "The detention center is the space where the anxieties surrounding mobility become crystallized and where the distinctions between citizen and other must be sustained in the minutiae of everyday life" (Hall, 2010, p. 883). Moreover in some regions of Mexico, the work of INM officers can be more affected by the context of criminal violence and the presence of organised crime, including their work sites, i.e. on the routes where they carry out checkpoints, transfers and in the places where some immigration stations are located. This risky environment adds to the insecurities and anxieties they face daily and leads to emotionally charged forms of punishment.

Every time I visit detention centres, I observe and try to absorb all the information I receive through gestures, conversations, and looks. This type of research is experienced emotionally (Arditti, Joest, Lambert-Shute, & Walker, 2010; Bergman Blix & Wettergren, 2015; Burkitt, 2011). It is easy to feel tension, sadness, anger, fear, frustration, disgust, and empathy. These emotions are part of daily interaction between all people: migrants, INM agents, security and cleaning personnel, activists, and representatives of international organizations. While this range of emotions characterises everyday interactions, discussing these issues with INM officers was difficult. The interviews in which I could talk about emotions were mostly with women. In the case of the men, it was instead through informal interactions as part of participant observations that I got closer to how they express their emotions with migrants.

As I will further explain, the research found both contradictions and nuances in the emotions expressed by INM agents. The same agent could be both indifferent and compassive, and express vulnerability and aggression, which is similar to Aliverti´s findings with border enforcement UK agents. "In literally putting their bodies on the line, they convey the emotionally and morally draining nature of border controls and its human costs on both sides of state coercion, which exercise can equally brutalize and humanize those bestowing it" (2021, p. 152). In this sense, emotions as an analytical lens of punitive practices in detention centres allow us to analyse punitive subjectivities of immigration agents.

**Punishing subjectivities**

When I arrived to the INM Regulation facilities in Ciudad Hidalgo (border with Guatemala) in March 2022, for three consecutive days migrant families mainly from Honduras, El Salvador, Cuba, Ecuador and Venezuela waited outside these offices without access to basic services and with no information from the government about their chances to get humanitarian visas. An INM agent guarding the facilities said this phrase to me.

*"These (addressing migrants) are like a tantrum child; what they want is for us to give them candy, but if you give candy to a tantrum child, they will want more and more; that is how these people are, but no way, the law is the law, and we have to respect it" (interaction with an INM agent, 2022).*

He was not only infantilizing migrants but justifying his acts as if a way of punishing or disciplining naughty children. This is one example of how the daily treatment that detained migrants receive is based on punishment. In a sense, the institutional practices in detention centres promote punitive subjectivities (Carvalho & Chamberlen, 2024) of their staff, which means that the expected attitude and treatment toward migrants is one of discipline and punishment, especially emotional punishment[6]. Moreover, just as the example above shows, the normative framework "[…] conditions individuals to feel emotional attachment to the legal norms, and to feel motivated to desire those who break these norms to be punished" (Carvalho & Chamberlen, 2016, p.11).

Punitive subjectivity is formed gradually by belonging to an institution characterised by a national security identity, specially in the context of criminal violence in Mexico and the criminalisation of migrants. INM agents work in insecure and precarious conditions, i.e. there are both internal and external factors that wear agents down emotionally, which facilitates the creation of punitive subjectivities. INM agents are an example of subjects who "[...] manage their feelings and insecurity and anxiety by producing or reinforcing illusions of order and control, at the same time as it provides these people with a channel through which to express their frustration by projecting hostile feelings towards criminalised others" (Carvalho & Chamberlen, 2024, p. 173). In a way, I argue that INM

---

[6] Although there is also evidence of physical punishment, and even migrant prisons are called torturing environments (GICDMT, 2018).

as an institution brings people together not simply as law-abiding members but as punishers. The following testimony is from a former female INM agent.

> *I had very young colleagues working with me at INM, and it was like feeling power or authority, because when you were with a state or municipal police officer you were like his boss; he had to listen to you. When you are young that intoxication of authority and power made you say, "I'm in charge" (Female former INM agent).*

Punishing is not only a way of feeling empowered, but from the institutional perspective it is a way of belonging and differentiating from potentially dangerous migrants. Therefore, punishment positions the punishers in the right side of order. The symbolic and real power of wearing a INM uniform make them belong to a national security institution. Depending on the region, working as an INM agent can be considered high-status. For example, in Tapachula, it is one of the most recognised jobs. However, the average salary is low, around 650 euros per month (Flores, 2023). There are different profiles; while some agents have very low skills, others are professional, committed, and have good academic and professional backgrounds. Unfortunately the type of community they belong to is one that is sustained through the punishment of migrants, and this has particular consequences, since "[…] the image of community and belonging that it produces is precisely one which depends on punishment for its maintenance" (Carvalho & Chamberlen, 2024, p. 171).

The functioning of a bureaucratic structure is a key element to understand the formation of punitive subjectivities. Immigration bureaucracies function based on the depersonalization (Ferguson, 1984) in a vertical structure that activates "[…] the potential for bureaucracies to diffuse individual responsibility for wrongdoing by elevating efficiency over ethical concerns and disguising the true nature of collective acts" (Weber, 2005, p. 91). The following testimony by an officer from the National Commission on Human Rights who was in charge of the training of INM agents shows how this institution works through processes that disassociates subjects from their moral choices, as well as routinization that hides consequences of their actions (Weber, 2005; Weber & Landman, 2002). Decisions are often taken in a routine and fragmented manner, which reduces responsibility their actions.

*As public servants, they have an obligation, and they must comply with that obligation. The problem is that they may end up doing things they ignore. For example, there have been several cases where INM agents are in judicial proceedings because they acted wrong while obeying their superiors. After all, another public servant in higher command told them, "You sign here" (interview with National Commission of Human Rights officer in charge of training INM agents, 2017).*

It is therefore essential to analyse the institutional contexts and working conditions of INM agents assigned to work in border areas, checkpoints, and detention centres. Within the INM, working conditions tend to be precarious, and power relations usually subjugate those at the lower levels of the hierarchy, who are even mistreated by their superiors. Just as the following testimony shows, this creates a scenario in which punishment and trauma may be part of the working conditions at INM.

*I was burnt out with the foreigners, listening to their stories. Moreover, it was emotional fatigue with my bosses, explaining why human rights should be respected. So many times, they would tell me: "You look like you come from the CNDH (Human Rights National Commission)." Finally, my boss would tell me: "You're an asshole; mind your own business" (interview with a former female agent of the INM, 2018).*

In my interviews with women officers, I identified gender inequalities at work, particularly around care work and family separation. Working hours are long, and it is common for them to be moved from city to city regularly, which may be more problematic for those who are mothers or caregivers since they are not fulfilling their traditional role as carers. I found that many officers are single mothers. So, in cases where they move to other cities, they need more support networks. I will return to these cases later, as the most common expressions of empathy come from women agents who are mothers.

*It is challenging, it is tough because I am not from here, and I had to come here, so it is complicated for me because of the schedules with my daughters, so I have no one here to take care of them. I would like to change my schedule. It is*

*supposed to be until six in the afternoon, but it's complicated (interview with a female INM agent, 2017).*

The lack of training and institutional support to acknowledge the responsibilities and obligations of being an INM agent can have severe consequences for guaranteeing the human rights of persons in immigration detention. Moreover, practices of exceptionality and discretion, together with the lack of training, generates less regulated environments for punishment.

*The public servant in Chiapas is exhausted. They often say "I work all the time, there are people all the time." Moreover, they are working overtime that they are not paid for. Sometimes they don´t really know what to do or generally what they were hired to do (interview with CNDH oficial in charge of training INM agents, 2017).*

I met officers, especially young ones, who wanted to carry out their work according to regulations, to provide services to persons in detention according to their needs, for example, in case they required translation services or dietary restrictions. This sometimes led to problems and mistreatment from their superiors and caused them even more work-related stress.

*The head of Control and Verification told me, "You know what, I can't deal with the Hindus anymore; nobody understands them at all" he said, "They don't want to eat, they don't bathe," because they were on a strike of not bathing and they smelled horrible after five or six days without bathing. So he told me: "Go talk to them." Moreover, when I went to tell them that they had the right to a lawyer and not to be deported, my boss told me, "How long are they going to last here? And they don't bathe, and we have to provide them special food". So he wanted to deport them, and I told him, "But you can't deport them," and he told me, "But why not" (interview with a former female INM agent, 2018).*

On the contrary, I had also known officers who embody a punitive subjectivity. They usually have been in the institution for a more extended period, often participate in

detention operations[7] or work under a lot of stress guarding spaces with few staff. These agents usually expect or demand gratitude and obedience from migrants and can become enraged when migrants demand recognition of their rights and access to justice.

I also observe that depending on the profile of the person in charge of the INM office or the immigration detention centre, labour dynamics can generate additional problems that affect work such as: constant fear of losing their jobs, burnout for not having rest hours or access to psychological support, working overtime without pay, sexism and racism in labour relations, which can even lead to labour and sexual harassment. In these contexts, INM offices become spaces where one can experience punishment as a trickle from above.

> *Yes, we have had extreme cases of violence against migrants, but we do not have a way to unburden ourselves. Nobody says: "Oh, we are going to support you with a psychologist or something, so that you can talk and feel relieved" (interview with INM female agent, 2018).*

Burnt out officers as well as those who embody a punitive subjectivity may punish migrants in different forms. Some punishments might appear to be more subtle or routinized as part of everyday work, such as denying a sanitary towel, stripping a person of his or her shoelaces, denying a phone call or a visit to the doctor. There are also the most extreme forms of punishment, such as cases of torture and State crimes like what happened in Ciudad Juarez on March 27, 2023, where 40 people died when the immigration centre caught fire and migrants where locked inside the place, because apparently the chief officer gave the order not to open the door.

**Punitive subjectivity through fear, disgust and empathy**
**A context of fear**

For Sara Ahmed, fear is an emotion that responds to something that is not yet present, that is, to the threat that something might happen. "[...] fear does not involve the defense of borders that already exist; rather, fear makes those borders, by establishing

---

[7] Not all immigration agents take part in detention operations. There are some INM agents, specially female who work doing paperwork or administrative tasks and have less  direct contact with migrants.

objects from which the subject, in fearing, can stand apart, objects that become 'the not' from which the subject appears to flee" (Ahmed, 2004, p. 128). Migration agents work in an environment that constantly instils fear. I observed that they are trained to distrust and fear. In my analysis, I identify distrust and fear from discourse and practice as central elements that constitute institutional dynamics and constantly create a difference between "us" and "them."

Both distrust and fear are elements present in their daily work that derive from two parallel processes that have taken place over the last two decades. In the first place, a growing criminalisation public discourse triggered by 9/11 that reinforces the idea of migrants as a threat to national security[8]. In this sense, although the INM is an agency whose functions fall under administrative law, it is a national security agency created in the early 90s with a militaristic approach or "[...] a set of beliefs, values, and assumptions that stress the use of force and threat of violence as the most appropriate and efficacious means to solve problems" (Kraska, 2007, p. 503). Although the regulations stipulate that their administrative duties ensure orderly, safe and regular migration, they are trained to identify and detain migrants.

> *The structure is organized from above, and sometimes I perceive that INM agents even have a particular fear of their superiors. Chief officers sell them the idea of "this is national security" (interview with CNDH oficial in charge of training INM agents, 2017).*

Secondly, the war against drug trafficking initiated by President Felipe Calderón in 2006 has generated a context of extreme criminal violence. Between 2007 and 2017, at least 213,000 men and 25,000 women were killed in Mexico (Data Cívica, & CIDE, 2019). Migrants are a highly vulnerable group because they are extorted, kidnapped and murdered[9]. Organised crime surveils some of the traditional migration routes close to the places where immigration detention operations are carried out. The presence of these

---

[8] During the six-year term of Enrique Peña Nieto, the *Plan de Seguridad Nacional 2014 - 2018* openly contemplated borders, seas, and irregular migration among the principal risks and threats to the country (Presidencia de la República, 2014).

[9] One of the most severe cases is the San Fernando Massacre that occurred on 22 August, 2010 in Tamaulipas, 150 kilometres from the US border. The victims were 72 migrants of various origins during their transit through Mexico on their way to the United States.

criminal groups generates risky contexts, which can also provoke fear among immigration agents. The following is a testimony of a former INM female agent.

> *They (INM agents) took photographs of this mole that I have, so if I had an accident and they found me dead, if there were only pieces of me, they could identify me (interview with a female former INM agent, 2017).*

In this testimony, the agent states that when she joined the INM, as part of the recruitment process, the agency had to ensure a way to identify her body in case of an accident. However, the second part of her quote demonstrates that the work context involved not only having an accident but also the possibility of her body being found dismembered. It is a testimony that shows the level of risk that INM agents must assume when hired.

Interaction with organised crime also happens in immigration detention centres. Although these situations are by no means generalisable, it does happen that members of criminal organisations such as *Mara Salvatrucha* from Guatemala, El Salvador or Honduras arrive in Mexico. In one of the testimonies, the INM agent narrates the fear she felt when interacting with these people and how the other agents subordinated themselves to the members of this group inside the detention centre.

> *I remember a Mara who used to grab the soccer goal, put it in the middle of the courtyard and hang from the highest part to do push-ups. He would do it shirtless and show all his tattoos, and if anyone would stare at him, he would challenge them with his eyes. I remember the agents telling me: "He is the one in charge now" (interview with a female former INM agent, 2018).*

Finally, there is a third element that instils fear, and has to do with an institutional culture of very precarious working conditions as I just mentioned before. There is a high turnover of personnel, and they perceive that their work is at risk if they do something wrong or against their superiors' will. In an interview in 2023 with a high-ranking INM official, she mentioned to me that: "INM agents are afraid of being fired; they are constantly threatened with the idea that they will lose their jobs".

**Disgust to keep distance and stay in the "good" side**

*"The difference is clear; when you enter the areas where they have them, the first thing you do instinctively when you cross the door to where the population is, let's say, is to do this (makes a gesture as if to vomit), I mean, I don't want to touch anything, right? So the rooms smell bad, they clean them, but for some reason, they smell bad, the bathrooms are communal, and they are not clean either"* (interview with a female former INM official, 2018).

The detention centres as well as detention settings I have visited in Chiapas (parks, car parks, roadside spaces and public spaces in the vicinity of INM offices), where migrants wait for days and sometimes weeks for immigration procedures, are characterised by the social neglect of the State. Unsanitary conditions are a constant in these spaces. There is no access to water or toilets. In 2023, I visited the facilities of the office of the *Secretaría de Bientestar* [Welfare Ministry] in Tapachula, where migrants came every day to participate in a temporary employment programme. Even in that large courtyard inside the offices, there were no toilets, and people had to defecate in a vacant lot in the open air. Again and again, the same image is repeated. That's why I started writing about the disgust. I have never been able to normalise the highly unsanitary conditions when I do fieldwork, whether inside a detention centre or in detention settings. The State, whether through the INM or municipal governments, does not consider guaranteeing the human right to access water and the dignity that anyone deserves to defecate and urinate in clean and private places.

Disgust is a cultural construct and a social emotion condemning an object or subject as a contaminant, usually related to waste and human body secretions (Nussbaum, 2004). It is also considered a feeling and emotion of self-protection and survival (Asselborn, 2012) produced by the belief of being contaminated by an object, situation or person considered polluting. As a cultural and social construct, each society and culture creates its own beliefs about contamination and disgust. However, from a Western perspective, despite the cultural variants that may exist, disgust is concentrated in a differentiation that prioritizes what is considered aesthetically beautiful, clean, and desirable, as opposed to what is considered aesthetically ugly, dirty, and disgusting.

The State is a central transmitter of messages about polluting subjects. Moreover, most societies teach the avoidance of particular groups of people as physically disgusting, bearers of contamination that the healthy element of society must keep at bay" (Nussbaum, 2004, p. 72). These are the social groups of the periphery, the categories of

the excluded, the lower social classes, the sick, the imprisoned, the homeless, and the migrants. The American anthropologist Mary Douglas (1973) speaks of pollution's instrumental and moral effect; she exposes a social and cultural order that finds its differentiation based on the belief about that which pollutes. In her famous phrase, "dirt offends order," she shows that the ideas of dirt and pollution are not isolated but part of a system that accepts or rejects, based on the classification of what may or may not be polluting objects.

Disgust is an essential human reaction to the abject. It is an emotion that separates and reminds us of our animality, thus making us claim to be cultural and "civilized" beings. "This disgust begins in the moral and social realms and goes on to take the form of smells, shrinks, and ugliness" (Miller, 1998, p. 348). The following testimony by a Honduran detainee at *Estación Migrartoria Siglo XXI* shows how disgust allows social and political distance, which are basic components of a punitive subjectivity; it makes INM agents believe they are morally more elevated than those considered contaminated, in this case a black Garifuna family isolated in this place.

*They didn't want to give them food, they didn't even take them out for dinner! So I went to talk to a security guard and said, "There's the food for the isolated girls, aren't they going to give it to them?" When I saw that, the officer came with gloves, with a mask, and I was like [...] What could these people have? Why are they coming, like this, with disgust? When she opened the cell, she brought the food to them, as if the guard was disgusted when she saw those people (Honduran migrant recounting an incident with Garifuna girls detained in Estación Migratoria Siglo XXI).*

Disgust becomes a powerful mechanism of rejection and punishment, which, together with racism, leads to humiliation and dehumanization practices. It is a form of punishment because people are forced to feel disgust of the environment and of themselves. Many times migrants in detention have asked me to bring them toothpaste or deodorant. One young man told me once he could not recognize himself in the mirror after three months of detention, he told me he felt dirty and abandoned. The unsanitary conditions, unpalatable or rancid food, limited access to sanitary napkins and diapers, lack of privacy, and overcrowding are experienced by migrants as a form of punishment that goes beyond discomfort.  They feel even more vulnerable to demand the fulfillment of their

rights because they do not even have a decent bathroom where they can wash, access to soap and clean water.

**Empathy within a punitive system**

In May 2019, my colleagues and I interviewed the person who, at the time, was in charge of the INM regulatory office in Ciudad Hidalgo. She was a woman with many years of experience who worked 12-hour days or more. During the interview, she showed herself as a responsible and committed official. She explained to us the various actions she had taken to ensure that people could wait for their procedures in more dignified conditions. At one point, I asked her about one of the most difficult moments she had experienced with the arrival of the Central American migrant caravans in 2018. She tearfully replied that she remembered very well that she had been at the gateway to Mexico in the first caravan and met several women with their children. She told them they could pass, but the male leaders said they had to wait so the whole group can enter together. She recalled one woman with two small, exhausted children whom she could not convince to enter before her companions. She said she arrived home that night, hugged her daughters and burst into tears.

Some INM guards show empathy with specific profiles of asylum seekers, even going so far as to understand the reasons and risks faced by migrants. Just as in the previous case, through interviews, but most important through the informal ethnographic interactions and observations, it is possible to identify moments where INM agents show this empathy, recognizing that detainees are similar to them.

> *It is difficult because they (the migrants) sometimes talk to us, even crying. So we prefer to inquire less if they have already made the complaint, we read the case before interviewing them, so we try not to ask them because it is traumatic for them to remember it (interview with female INM agent, 2017).*

> *I like to support people; I don't find it difficult; I never give them extra problems. For example, if I see that the card is damaged, I find a solution. They have a certificate or something, but I don't give them more troubles; I help them with their process because I understand it. If I go to another institution, I want them to serve me well (interview with female INM agent, 2017).*

Although I have identified male agents who show empathy and ways of helping migrants, I note that it is more common to observe empathy in female agents. Both motherhood and domestic violence stand out as situations in which INM agents tend to be much more empathetic in supporting migrant women.

*I gave her my card (to the detained immigrant) and told her: "Don't go back to your husband; the next time, he will kill you. If you want, I can help you, we will get you to a shelter, and we will get your papers done". (interview with a female former INM official, 2018).*

In the following testimony, a Honduran migrant woman who was detained for three weeks with her children, a three-year-old boy and a nine-month-old baby girl, at the *Estación Migratoria Siglo XXI* explained to me that she got along well with some of the guards, especially the ones who were mothers. She established a certain closeness with some of them, allowing her to negotiate better conditions than other women in detention.

*That guard was removed from one station to another; she had been there for over two months. One day my child fell and hit his forehead. She started crying. I told her: "calm down". She cried when she saw my son crying because he reminded her of her children. She had been unable to see her children for over two months (Honduran migrant woman detained in Estación Migratoria Siglo XXI).*

Just like in the opening vignette, in very different circumstances these two women faced stress in caring for their children. Displays of empathy in the institutional environment of repression and dehumanization that comes with immigration detention generate a fragile kind of empathy (Hall, 2010). This emotion runs counter to institutional practices that seek social distance between migrants and INM agents. Therefore, an excess of empathy could even be frowned upon or punished by one's peers. During one of my visits, there was a very sick Haitian young male outside the immigration detention centre in Tapachula. It was midday, and it was very hot, around 37ºC. This boy was laying under a tree with his family, and no one had called an ambulance. As we were leaving the place, an INM agent called me very discreetly and told me that there was a young man with a medical emergency; he pointed me to where he was. I understood that if the agent

called an ambulance or alerted his colleagues, it could be frowned upon or have negative consequences for him. After almost an hour, we managed to get the paramedics of the *Grupo Beta* of the INM to take the young man to a hospital. Unfortunately, four hours later, they returned him to the same place, arguing that the hospital could not treat him due to a lack of medication.

The documented gestures and actions show that there are people within the INM who do not forge a punitive subjectivity. They can exhibit contradictions by being indifferent at certain times and empathetic at others. To be compassionate, they must take risks, such as being observed and scolded by their superior. They are people who, like Nancy, can end up physically and emotionally affected and who, in many cases, make a difference and provides support to access essential services in detention. These are the subjectivities that resist the punitive environment.

**Conclusions**

Punitive environments begin with the public discourse that criminalises migrants and legal frameworks that provide space for exceptionality and punishment. Even though irregular migration refers to administrative law, it is an issue politically and practically managed as national security. Therefore, INM institutional belonging requires the development of a punitive subjectivity. In these terms, many INM agents experience first-hand the punishment or potential punishment by their superiors. Once they become members, they learn that the institution demands an attitude of discipline and punishment among them, and especially towards migrants. Therefore, a central element to analyse is how the institution reinforces itself through punishment. In addition, the context of criminal violence in Mexico, plus the precariousness of employment, generates environments of risk and insecurity that feed punitive environments.

Emotions such as fear and disgust, along with other emotions that I do not address in this paper, such as anger, antipathy and indifference, reinforce the punitive subjectivities of the agents. On the contrary, compassion and empathy confront them. Without generalising this finding, as male officers have shown empathy on certain occasions, women tend to be more empathetic than men. This is possible because they share experiences of gender inequality in their family, community and work environments,

especially violence and abuse with migrant women, as well as care and mothering practices.

In this article, I proved that emotions are critical elements in forming punitive subjectivities. I also highlighted that institutions must be studied from the inside and the members' perspectives, therefore, the value of institutional ethnography. This research allowed me to acknowledge the centrality of emotions in the working conditions and daily interactions of INM officers. I demonstrated that punishment is a practice of belonging and reinforcing the institution's value and that provides INM agents with a way to channel emotions derived from the anxieties and frustrations of their working environment. I finally explained, through the cases of empathy, that punitive subjectivities are not fixed; on the contrary, its emotional dimension, in some instances, allows space for humanization and care.

## Sources

Ahmed, S. (2004). Affective Economies. *Social Text*, *22*(2), 117–139.

Aliverti, A. (2021). *Policing the Borders Within*. Oxford University Press.

Araiza, O., Buttery, H., Rossi, V., & Spalding, S. (2019). *La Implementación y el Legado del Programa Frontera Sur de México*. (S. Leutert, Ed.). Austin: LBJ School The University of Texas, Austin, Robert Strauss Center for International Security and Law, FM4 Paso Libre, El Colegio de la Frontera Norte.

Arditti, J., Joest, K., Lambert-Shute, J., & Walker, L. (2010). The Role of Emotions in Fieldwork: A Self-Study of Family Research in a Corrections Setting. *The Qualitative Report*, *15*(6), 1387–1414.

Ariza, M. (2016). La sociología de las emociones como plataforma para la investigación social. In M. Ariza (Ed.), *Emociones, Afectos y Sociología Diálogos desde la investigación social y la interdisciplina* (pp. 7–35). Cd. de México: Instituto de Investigaciones Sociales, UNAM.

Ariza, M. (2021). The Sociology of Emotions in Latin America. *Annual Review of Sociology*, *47*, 157–175.

Benett, J. (2016). *The Working Lives of Prison Managers. Global Change, Local Culture and Individual Agency in the Late Modern Prison*. Palgrave Macmillan.

Bergman Blix, S., & Wettergren, Å. (2015). The emotional labour of gaining and maintaining access to the field. *Qualitative Research*, *15*(6), 688–704.

Bilgic, A., & Gkouti, A. (2021). Who is entitled to feel in the age of populism? Women´s resistance to migrant detention in Britain. *International Affairs*, *97*(2), 483–502.

Burkitt, I. (2011). Emotional Reflexivity: Feeling, Emotion and Imagination in Reflexive Dialogues. *Sociology*, *46*(3), 458–472.

Carvalho, H., & Chamberlen, A. (2016). Punishment, Justice, and Emotions. In *Oxford Handbooks Online: Criminology and Criminal Justice*. Oxford University Press. https://doi.org/10.1093/oxfordhb/9780199935383.013.138

Carvalho, H., Chamberlen, A., & Lewis, R. (2020). Punitiveness beyond Criminal Justice: Punishable and Punitive Subjects in an Era of Prevention, Anti- Migration and Austerity. *The British Journal of Criminology*, *60*(2), 265–284.

Carvalho, H., & Chamberlen, A. (2024). *Questioning Punishment*. Routledge.

Ceja Cárdenas, I., & Miranda, B. (2022, May). La espera como técnica de gobierno de las migraciones en las Américas. *Revista Común*.

Cívica, D., & CIDE. (2019). *Claves para entender y prevenir los asesinatos de mujeres en México*.

CPDTMF. (2021). Observación general núm. 5 (2020), sobre los derechos de los migrantes a la libertad y a la protección contra la detención arbitraria. Comité de Protección de los Derechos de Todos los Trabajadores Migratorios y de sus Familiares.

Crawford, N. (2014). Institutionalizing passion in world politics: fear and empathy. In *Forum: Emotions and World Politics* (pp. 535–557). Cambridge: Cambridge University Press. https://doi.org/https://doi.org/10.1017/S1752971914000256

Crawley, E. (2013). *Doing Prison Work. The public and private lives of prison officers* (Second). Routledge.

Douglas, M. (1973). *Pureza y Peligro. Un análisis de los conceptos de contaminación y tabú* (Primera en). Ciudad de México: Siglo XXI.

Ferguson, K. (1984). *The feminist case against bureaucracy*. (R. Steinberg, Ed.). Philadelphia: Temple University Press.

Fernández de la Reguera, A. (2020). *Detención migratoria. Prácticas de humillación, asco y desprecio.* Ciudad de México: UNAM.

Fernández de la Reguera, A. (2022). Immigration Detention, the Patriarchal State and the Politics of Disgust in the Hands of Street-level Bureaucrats. *Feminist Encounters: A Journal of Critical Studies in Culture and Politics*, *6*(2), 30. https://doi.org/https://doi.org/10.20897/femenc/12353

Flores, C. (2023, December 15). ¿Cuánto gana un agente de migración en México? Éste es el sueldo de un . *Infobae*.

GICDMT. (2018). *Detención migratoria y tortura: Del Estado de Excepción al Estado de Derecho.*

Global Detention, P. (2021). *Immigration Detention in Mexico: Between the United States and Central America*. *Global Detention Project Country Profile*.

Graham, M. (2002). Emotional Bureaucracies: Emotions, Civil Servants, and Immigrants in the Swedish Welfare State. *Ethos*, *30*(3), 199–226.

Griffiths, M. (2023). The emotional governance of immigration controls. *Identities*, 1–22. https://doi.org/10.1080/1070289X.2023.2257957

Hall, A. (2010). `These People Could Be Anyone´: Fear, Contempt (and Empathy) in a British Immigration Removal Centre. *Journal of Ethnic and Migration Studies*, *36*(6), 881–898.

Hall, A. (2012). *Boder Watch Cultures of Immigration, Detention and Control*. New York: PlutoPress.

Herrera, J. M., & Rivera, M. (2021). Migration, emotions and policies of sensibilities in Central America. *International Sociology*, *36*(4), 569–584.

Herzfeld, M. (1992). *The social production of indifference. Exploring the symbolic roots of western bureaucracy*. Chicago: The University of Chicago Press.

Kearney, G., Corman, M., Hart, N., Johnston, J., & Gormley, G. (2019). Why institutional ethnography? Why now? Institutional ethnography in health professions education. *Perspectives on Medical Education*, *8*, 17–24.

Kelman, H. C., & Hamilton, V. L. (1989). *Crimes of Obedience: Toward a Social Psychology of Authority and Obedience*. London: Yale University Press.

Kemper, T. (2006). Power and Status and the Power-Status Theory of Emotions. In J. Stets & J. Turner (Eds.), *Handbook of the Sociology of Emotions* (pp. 87–113). New York: Springer.

Kraska, P. (2007). Militarization and Policing - Its Relevance to 21st Century Police. *Policing*, *1*(4), 501–513.

Meier, I. (2020). Affective border violence: Mapping everyday asylum precarities across different spaces and temporalities. *Emotion, Space and Society*, *37*. https://doi.org/https://doi.org/10.1016/j.emopsa.2020.100702

Menjívar, C., & Abrego, L. (2012). Legal Violence: Immigration Law and the Lives of Central American Immigrants. *American Journal of Sociology*, *117*(5), 1380–1421.

Miller, I. (1998). *La anatomía del asco*. Madrid: Taurus Pensamiento.

Miranda, B. (2023). Migración africana en situación de espera: nuevo alcance y dimensión de la contención migratoria en México. *Revista Pueblos y Fronteras*, *18*.

Moss, P., & Prince, M. (2017). Helping traumatized warrios: Mobilizing emotions, unsettling orders. *Emotion, Space and Society*, *24*, 57–65.

Mountz, A. (2017). Island detention: Affective eruption as trauma´s disruption. *Emotion, Space and Society*, *24*, 74–82.

Nussbaum, M. (2004). *Hiding from Humanity Disgust, Shame and the Law*. Princeton: Princeton University Press.

Presthus, R. (1958). Toward a Theory of Organizational Behavior. *Administrative Science Quarterly*, *3*(1), 48–72.

Puthoopparambil, S., Ahlberg, B., & Bjerneld, M. (2015). It is a Thin Line to Walk on: Challenges of Staff Working at Swedish Immigration Detention Centres. *International Journal of Qualitative Studies on Health and Well-Being*, *10*, 1–11.

República, P. de la. (2014). *Programa para la Seguridad Nacional 2014 - 2018. Una política multidimensional para México en el siglo XXI*. Ciudad de México.

Savio Vammen, I. M., & Syppli Kohl, K. (2022). Affective Borderwork: Governance of Unwanted Migration to Europe Through Emotions. *Journal of Borderlands Studies*.

Sim, J. (2012). An inconvenient criminological truth: pain, punishment and prison officers. In J. Bennett, B. Crewe, & A. Wahidin (Eds.), *Understanding Prison Staff* (Second). Routledge.

Smith, D. (2001). Texts and the Ontology of Organizations and Institutions. *Studies in Cultures, Organizations, and Societies*, *7*, 159–198.

Tessitore, F., Caffieri, A., Parola, A., Cozzolino, M., & Margherita, G. (2023). The Role of Emotion Regulation as a Potential Mediator between Secondary Traumatic Stress, Burnout, and Compassion Satisfaction in Professionals Working in the Forced Migration Field. *International Journal of Environmental Research and Public Health*, *20*(2266).

Turner, J., & Stets, J. (2006). Moral Emotions. In J. Stets & J. Turner (Eds.), *Handbook of the Sociology of Emotions* (pp. 544–566). New York: Springer.

Unidad de Política Migratoria, S. (2021). *Boletín Mensual de Estadísticas Migratorias*.

Unidad de Política Migratoria, S. (2022). *Estadísticas migratorias. Síntesis 2022*.

Unidad de Política Migratoria, S. (2023). *Boletín Mensual de Estadísticas Migratorias*.

Vega, I. (2017). Empathy, morality, and criminality: the legitimation narratives of U.S. Border Patrol agents. *Journal of Ethnic and Migration Studies*, *44*(15), 2544–2561.

Weber, L. (2005). The Detention of Asylum Seekers as a Crime of Obedience. *Critical Criminology*, *13*, 89–109.

Weber, L., & Landman, T. (2002). *Deciding to Detain The organisational context for decisions to detain asylum seekers at UK ports*. Essex.

Wettergren, Å. (2010). Managing unlawful feelings: the emotional regime of the Swedish migration board. *International Journal Work Organisation and Emotion*, *3*(4), 400–

419.