# Peer review of "Punitive subjectivities and emotions in immigration detention"

_Migration Politics_

## Round 1 · Referee Report · Anonymous · 2024-4-2

Strengths

1. Examines the underexplored case of migration control in Mexico.

2. Focuses on agents of control's perspective adding to the literature on border bureaucracies.

3. Adds to the growing literature on the role of emotions and emotional interactions in shaping interactions in punitive environments.

Weaknesses

1. The literature review requires more depth on (i) migration bureaucracies and (ii) on the case study.

2. The expanded literature review will allow for a more in-depth analysis in the results sections and give greater conceptual emphasis to the empirical data.

3. No details on ethical protocol are provided in relation to this study.

Report

The article would benefit greatly from the literature on the border as practice and border bureaucracies. This would allow the author to strengthen her engagement with the question "Under what structural and institutional conditions do emotions become power in immigration detention settings?" as well as broaden her analysis of agents' discretionality. Overall, the review of these bodies of literature will allow the author to situate more clearly the conceptual contribution of the paper beyond the empirical one, which is very evident.

The author stresses that “little is known about the operation of migration detention in Mexico” (page 4). While it is true that research on this front is scarce, peer-reviewed papers on this topic have been published in recent years and it would be valuable for the author to include them, as this would also make it possible to situate her work in relation to (and in conversation with) the research that accounts for this case study.

The author situates her methodological approach within the framework of institutional ethnography and her fieldwork linked to her participation in a Human Rights Centre. She gives valuable insights into her use of participatory observation as a method of data collection, however, some details about the ethical protocol surrounding the study are omitted.

Finally, I was puzzled by the author's decision to include the nationality of the authors referred to in the article. I wonder whether this is a criticism or an expression of methodological nationalism.

Requested changes

1. Engage with the literature on Bordering as a practice and Border bureaucracies. Check for example the work of Karine Côté-Boucher, Barak Kalir, Christin Acherman, and Lisa Borrelli.

2. Strengthen the literature review about the case study. In particular, four papers are critical; further, the last one could help the author to broaden the discussion on the empathy section: (1) Abnormal bordering: control, punishment and deterrence in Mexico’s migrant detention centres; (2) Torturing environments and multiple injuries in Mexican migration detention; (3) Mexican bureaucrats and the everyday restriction of transnational migration in a context of scarcity; (4) Tactics of Empathy: The Intimate Geopolitics of Mexican Migrant Detention.

3. Expand reflection on the author's positionality and ethical considerations related to the study. For example, what were the ethical dilemmas and challenges related to her access to the detention centre as part of the psycho-legal accompaniment team, how the interviews with migrants differed from the accompaniment provided, how consent was given, how privacy was guaranteed, if applicable.

---

## Round 1 · Referee Report · Anonymous · 2024-5-6

Strengths

1.This paper documents the formation of punitive subjectivity through emotions among INM (Mexico) agents. This is a novel research topic and case study.
2. It draws on interviews and participant observation to identify the formation of emotional responses (fear, disgust, and empathy) and how it shapes guard subjectivity, i.e,. attitudes and orientations. This is difficult, painstaking work, and provides a new and useful lens to understand dynamics of detention and policy.
3. The subject responses are engaging.

Weaknesses

1. It was difficult to discern what the paper was about. I believe the best (and most accurate) version of the paper's key finding first appears on p. 24: "there are people within the INM who do not forge a punitive subjectivity. They can exhibit contradictions by being indifferent at certain times and empathetic at others." If this statement were upfront and in the intro (along with how you collect and analyze your data), the reader will have an easier time navigating the paper, which does a lot of things.
2. The paper needs some structure. Because it does so many things, giving the reader an outlining paragraph in the intro will help (1. why analyze emotions, 2. the role of emotions in shaping punitive subjectivity, 3. the case of INM, 4. evidence, 5. conclusion).
3. The writing was a little impersonal at times (e.g., "these stories made me realize"). This may be a style among institutional ethnographic work, but I think the analysis can be made without.
4. The section on emotions (fear, disgust, empathy) come after punitive subjectivity. Why? Do emotions shape subjectivity? Or does subjectivity make individuals conducive to some emotions versus others. Clarifying this will help 1. order the argument and 2. clarify it for the reader, both in terms of following the paper structure and understanding the relationship between concepts
5. as the paper is currently written, empathy comes as a surprise and feels out of place. this should not be the case. a strong and structured introduction will set the reader up to anticipate this shift. one idea was to move the motivating story (currently in the intro) to the analysis part of the paper.
6. I would suggest a stronger conclusion that addresses why attention to emotions is important. what do we understand differently about incarceration/detention or immigration as a result?

Report

Yes, this meets the journals criteria.

Requested changes

1. Edit for style and structure
2. Clarify argument up front, including signposting for the reader what the order of the paper will be
3. Stronger statement on the role of emotions in relation to creating or responding to subjectivity

Recommendation

Ask for minor revision

---

## Round 2 · Author Response

I am grateful for the revisions made to the first draft. I have worked with each of the comments and suggestions to improve the structure and clarity of the text. I intend to highlight not only the empirical contribution of the research but also the theoretical and methodological contribution to the field of border criminologies from the perspective of emotions as a source of knowledge to study forms of punishment and punitive subjectivities in immigration detention centres.

---

## Round 2 · List of Changes

1. I restructured the document to make the text clearer. The article now has 5 sections, as I integrated the methodological section at the beginning and modified the order so that the section on punitive subjectivities comes after the section on the analysis of fear, disgust and empathy.

  2. I engaged with the literature on Bordering as a practice and Border bureaucracies, specifically the work of Karine Côté-Boucher, Christin Acherman, and Lisa Borrelli.

  3. I also integrated two recently published texts on borderwork, emotions and institutions. (Marina Ariza 2024 y Billy Holzberg 2024).

  4. I strengthened the literature review about the case study. I added significant contributions of four papers based on detention centres in Mexico and specifically in Tapachula: (1) Abnormal bordering: control, punishment and deterrence in Mexico’s migrant detention centres; (2) Torturing environments and multiple injuries in Mexican migration detention; (3) Mexican bureaucrats and the everyday restriction of transnational migration in a context of scarcity; (4) Tactics of Empathy: The Intimate Geopolitics of Mexican Migrant Detention.

  5. I clarified some of the main findings of the paper which are: “There are people within the INM who do not forge a punitive subjectivity” ; and “ They can exhibit contradictions by being indifferent at certain times and empathetic at others”.

  6. I expanded a reflection on ethical considerations related to the differences between interviews with migrants in the frame of psycho-legal accompaniment and interviews for this research.

  7. I strengthened the conclusions.

  8. I edited style and structure.

  9. I removed the nationality of the cited authors to avoid misunderstandings on methodological nationalisms.

---

## Editorial Decision

unknown